# Rumen Fermentation of Feed Mixtures Supplemented with Clay Minerals in a Semicontinuous In Vitro System

**DOI:** 10.3390/ani12030345

**Published:** 2022-01-31

**Authors:** Zahia Amanzougarene, Manuel Fondevila

**Affiliations:** Departamento de Producción Animal y Ciencia de los Alimentos, Instituto Agroalimentario de Aragón (IA2), Universidad de Zaragoza-CITA, Miguel Servet 177, 50013 Zaragoza, Spain; zahiaagro@yahoo.fr

**Keywords:** zeolite, bentonite, sepiolite, microbial fermentation, in vitro semicontinuous culture system, medium pH

## Abstract

**Simple Summary:**

Mineral clays are included in the diets of ruminants to maintain health and improve productive performances. The inclusion of several types of mineral clay (zeolite, Z; bentonite, B; and sepiolite, S) in diets with different concentrate-to-forage proportions (65:35, HC, and 35:65, HF) was tested in vitro. In HC diets, the effect of Z manifested in a higher pH in the first part of fermentation, which can be related to a more stable rumen environment. The extent of substrate fermentation was lowest with S when added to the HC diet but was lowest with B when added to the HF diet. The response of the rumen environmental conditions and the extent of fermentation depends on the interaction between the type of clay and the proportion of concentrate and forage in an animal’s diet.

**Abstract:**

Interest in using clays in the diets of ruminants to improve health and performance is increasing. The microbial fermentation of 65:35 (HC) or 35:65 (HF) concentrate:forage feeds, alone or with zeolite (Z), bentonite (B), or sepiolite (S), was studied in an in vitro semicontinuous culture system. The medium pH was allowed to drop for the first 6 h and was gradually buffered thereafter. For the HC diet, the medium pH was higher with Z throughout incubation (*p* < 0.05). Similar results were observed for the HF diet, but with lower differences between the additives. Throughout incubation, the volume of gas produced was higher with HC than HF (*p* < 0.05). The gas volume with S was the lowest (*p* < 0.05) for HC, whereas for HF it was lowest with B from 8 h onwards (*p* < 0.05). No treatment differences (*p* > 0.05) were observed in dry matter disappearance, microbial mass, or volatile fatty acids. However, the inclusion of B in HC reduced the ammonia concentration at 6 and 12 h with respect to C (*p* < 0.05). The inclusion of zeolite as an additive in the diets of ruminants stabilizes the rumen environment during the first stages of fermentation in terms of pH and ammonia concentration, especially in high-concentrate diets. The buffering effect of bentonite and sepiolite was lower, and both might reduce ruminal microbial fermentation, depending on the concentrate proportion.

## 1. Introduction

Several clays are commonly used as technological additives in feed manufacturing as binder and anticaking agents [1], as well as mycotoxin binders [2,3], not only for improving feed quality, but also for enhancing the nutritive value of animal diets. Moreover, in ruminants, these natural feed additives have been proposed as buffers and osmoregulators to minimize the risk of impaired rumen fermentation promoted by high-energy, grain-based diets that may cause digestive disorders that would negatively affect animal health and productivity [4,5,6]. Among these clays, the most widely used are natural zeolite (clinoptilolite), bentonite, and sepiolite.

Clinoptilolite is a common form of natural zeolite. Zeolites are crystalline aluminosilicates with molecular dimensional pores which form larger molecular sieves. Several studies have shown the positive effect of supplementing cow diets with natural zeolites (clinoptilolite) in terms of rumen fermentation, increasing the pH, and the regulation of osmotic pressure [5,7,8]. Khachlouf et al. [9] state that this clay is also able to protect against mycotoxins, and for this purpose can be included at levels from 1 to 5 g/kg. In contrast, some studies have concluded that it has negative or no effects on milk yield, and the lower milk performance has been associated with a decrease in intake and digestibility [10,11].

Bentonite is a heterogeneous rock formed of highly colloidal and plastic clays composed mainly of montmorillonite [3]. Bentonite has been generally used as a technological additive in all feeds for all animal species and is also authorized for the reduction of feed contamination by mycotoxins [12]. Bentonite is the only technological additive registered by EFSA as a mycotoxin binder, at a dose of 20 g/kg. Furthermore, this natural clay can also be used for other beneficial purposes, such as to provide a buffering effect, preventing a drop in rumen pH during starch fermentation [4,6] and decreasing the rumen ammonia concentration to improve the feed and bacterial protein flow to the small intestine of sheep [13]. Bentonite may also have a role in trace element solubility in the rumen [14]. Positive responses in milk yield have been observed when bentonite is added to diets for lactating cows, mostly in diets including high concentrate proportions [15,16]. In contrast, the addition of bentonite has not shown a noticeable effect on buffering capacity, nitrogen utilization, or milk productivity when a high forage diet was fed [17,18,19]. 

Sepiolite is a clay mineral with a high porosity and surface area, strong absorptive power, high structural stability, and chemical inertia [20]. Sepiolite is also authorized to be added up to 20 g/kg to improve the physical characteristics of feeds. However, this clay is also used as a dietary additive in animal feeds for reducing the rate of passage through the gastrointestinal tract, thus allowing for a more efficient digestion [21]. Additionally, Fonty et al. [22] found that minerals contained in sepiolite contribute to rumen buffering, although Elitok and Guvlu [23] did not observe significant differences in rumen pH with or without sepiolite. Ivan et al. [24] reported that sepiolite increases the volatile fatty acid (VFA) concentration, and it could also reduce the number of bacteria and lead to lower methane production in the rumen. Moreover, Jouany and Morgavi [25] stated that sepiolite decreases rumen ammonia, carbon dioxide, and methane production. 

The potential response to each type of clay added to ruminant diets may vary depending on the rumen environmental conditions, which are modulated by the proportion of concentrate to forage. However, few studies have been carried out in this area. Therefore, the objective of this study was to contrast the effect of the inclusion of three types of clay (zeolite, bentonite, and sepiolite) on the rumen fermentation of either high-forage or high-concentrate mixed diets for milking cows. To achieve this, a semicontinuous in vitro system [26,27] was applied.

## 2. Materials and Methods

Four ingredients, namely, maize, barley, soybean meal, and wheat bran, were mixed at proportions of, respectively, 0:30, 0:30, 0:30, and 0:10 to simulate a standard concentrate feed used in diets for ruminants. This concentrate mixture was combined with alfalfa hay as forage source at either 65:35 or 35:65 concentrate-to-forage proportions, resulting in two experimental feed mixture diets for milking cows: high-concentrate (HC) and high-forage (HF). All ingredients (Table 1) were ground through a 1 mm sieve using a hammer mill (Retsch Gmbh/SK1/417449). Feed mixtures were supplemented with three different clay sources (zeolite: Z, 70–85% purity, 0–1 mm size; bentonite: B, 76.5% purity, <0.15 mm size; and sepiolite: S, 89.4% purity, <0.045 mm size), which were added at a proportion of 10 mg/g of total substrate. Clays were kindly provided by GORDES Zeolite AS. Unsupplemented high-concentrate (C) and high-forage (F) mixtures were also included as controls, resulting in four treatments per substrate mixture.

### 2.1. Experimental Procedures

To determine the fermentation kinetics, four in vitro incubation series were carried out for each forage-to-concentrate ratio, using a semicontinuous system [26] as modified by Prates et al. [27], with duplicate Erlenmeyer flasks (100 mL) per treatment. Rumen fluid obtained from three adult cannulated Rasa Aragonesa ewes fed on a 50:50 concentrate:forage mixture was sampled before the morning feeding, mixed, filtered through four layers of gauze, and used as inoculum. Ewes were housed in the facilities of the Servicio de Apoyo a la Experimentación Animal of the University of Zaragoza. Management and extraction procedures of rumen inoculum from donor animals were approved by the Ethics Committee for Animal Experimentation (procedure PI48/20). Care and management of animals agreed with the Spanish Policy for Animal Protection RD 53/2013, which complies with EU Directive 2010/63 on the protection of animals used for experimental and other scientific purposes.

Substrate samples (800 mg) were dispensed in 4 cm × 4 cm nylon bags (45 µm pore size, SEFAR-MAYSA, Barcelona, Spain), which were sealed and inserted into each flask. These flasks were filled under a CO_2_ flux with 80 mL of incubation solution [28], including 16 mL inoculum (0.20 of total volume). Normal rumen conditions were simulated by enabling a wide range of incubation pH, which was allowed to drop to around 6.0 in the first 8 h of incubation and then rise to around 6.5 from 8 h onwards. This was achieved by using two different CO_3_H^−^ concentrations (6 mM and 58 mM, respectively) when preparing the incubation solution [29] for these periods. Liquid outflow rate was fixed at 0.08 per hour by replacing 6.4 mL/h of incubation medium with the same volume of buffer solution without inoculum. Discrete liquid interchange was carried out manually every two hours from 0 to 12 h incubation and every 4 h from 12 to 24 h. Four 24 h incubation series were performed in a water bath at 39 °C. 

Gas production and incubation pH were recorded just before the liquid interchange, every 2 h in the first 12 h and every 4 h in the last 12 h, with a HD8804 manometer provided with a TP804 pressure gauge (DELTA OHM, Caselle di Selvazzano, Italy). Readings were corrected for atmospheric pressure and then converted to volume (mL) using a pre-established linear regression (*n* = 48, R^2^ = 0.993) and expressed per unit of incubated organic matter (OM). Samples of liquid medium (2 mL) were taken at 6 and 12 h and immediately frozen (−20 °C) for determination of the concentration of ammonia and VFA. Samples for ammonia analysis were taken on 2 mL of 0.2 N HCl, and those for VFA analysis were taken on 0.5 mL of a solution of 0.5 M phosphoric acid with 1 mg of 4-methyl valeric acid as internal standard. Aliquot samples of liquid medium were taken from liquid output of one repetition (flask) per treatment at every interval up to a total volume of 30 mL, to obtain a representative pool for the determination of microbial mass associated with the liquid fraction. Furthermore, at the end of the incubation period, substrate bags from the same flask were removed for determination of microbial mass associated with the solid fraction. These samples of liquid and solid fractions were frozen at −80 °C and lyophilized until analysis of bacterial concentration. The incubated bag with solid residue for the other flask on each treatment repetition was rinsed and dried at 60 °C for 48 h for the determination of dry matter disappearance (DMd).

### 2.2. Chemical and Microbiological Analyses

Substrates were analyzed following the procedures of AOAC [30] for DM (method ref. 934.01), organic matter (OM, ref. 942.05), CP (ref. 976.05), and EE (ref. 2003.05). Concentration of NDF was analyzed as described by Mertens [31] in an Ankom 200 Fibre Analyser (Ankom Technology, New York, NY, USA), using α-amylase and sodium sulphite, and results were expressed exclusive of residual ashes. Frozen samples of incubation medium were thawed and centrifuged at 13,000× *g* for 15 min at 4 °C, and the concentration of individual VFA in incubated medium was determined by gas chromatography on an Agilent 6890 apparatus (Agilent Technologies España S.L., Madrid, Spain) equipped with a capillary column (HP-FFAP Polyethylene glycol TPA, 30 m × 530 µm id × 1 µm film thickness). Concentration of ammonia was determined colorimetrically following the procedure of Chaney and Marbach [32]. Microbial concentration in lyophilized samples of liquid and solid media was determined using diaminopimelic acid (DAPA) as a marker, after hydrolysis in 6 N HCl for 12 h at 110 °C. Concentration of DAPA was analyzed by a combination of Ultrahigh-Performance Liquid Chromatography (UPLC, Waters, Milford, MA, USA) and mass spectrometry (Micromass MS Technologies, Manchester, UK), as in Guo et al. [33] for amino acid quantification. For use as standards, the microbial populations associated with the liquid (LAM) or the solid (SAM) fractions were extracted from rumen contents by cooling to 4 °C overnight in a methylcellulose solution [34]. Results of LAM and SAM were expressed per incubation volume or per unit of weight of incubated residue, respectively.

### 2.3. Statistical Analysis

Results were analyzed by ANOVA as a split-plot design, using the Statistix 10 package [35]. The incubation series (*n* = 4) was considered as a block, the effect of the substrate mixture (*n* = 2) as a main plot, and the additive (*n* = 4) as a subplot. The average of the two flasks for each experimental treatment within each incubation series was considered as the experimental unit. Differences were considered significant when *p* < 0.05. The Tukey *t*-test was applied at a *p* < 0.05 for multiple mean comparisons among treatments.

## 3. Results

The mean pH of the rumen inoculum at the start of the incubation series was 6.62 ± 0.16. As the incubation proceeded, the pH dropped to minimum values, resulting in average values of 6.08 and 6.04 at 8 h for the HC and HF mixtures, respectively. Thereafter, the pH was allowed to recover, reaching the maximum incubation pH at 20 h for both feed mixtures (6.62 and 6.57 for HC and HF, respectively). The differences recorded between the substrate mixtures from 2 to 10 h and at 16 h incubation (*p* < 0.05) were in all cases below 0.08 pH units. Since the interaction between the incubated substrate and the additive was significant at every point in time during the incubation, the results are presented separately for the HC and HF mixtures. With HC (Figure 1), the addition of Z recorded the highest incubation pH throughout the entire incubation period, the differences compared with the other treatments ranging from 0.06 to 0.15 pH units (*p* < 0.05). Furthermore, the pH was higher with B (*p* < 0.05) than C at 2, 6, and 8 h incubation. From 10 to 16 h incubation, differences between B, S, and C were not observed, but at 20 h S recorded a lower pH than B (*p* < 0.05). Less significant differences between additives were recorded with HF as the substrate (Figure 2), the pH with Z being higher than F at 6, 8, and 16 h, and a higher pH than F was also observed with B at 6 and 16 h and S at 16 h (*p* < 0.05). 

The volume of gas produced from fermentation was higher (*p* < 0.01) with HC than HF throughout the incubation period, being on average 133 and 99 mL/g OM, respectively, after 24 h of incubation (Table 2). As for the pH results, the comparison between the additives regarding the gas production pattern is shown separately for each substrate mixture. With HC, S recorded the lowest gas volume during the whole incubation period (*p* < 0.05). No differences were recorded at any time between C and B (*p* > 0.05), but Z produced a lower volume (*p* < 0.05) than B after 20 h and C at 24 h. When HF was the substrate mixture, the volume of gas with B was lower than with F and S at 4 and 6 h, and it was the lowest from 8 h onwards (*p* < 0.05). Gas production with Z was lower than with F in the first 4 h, whereas from 8 h onwards no differences (*p* > 0.05) were detected between S, F, and Z. 

No treatment differences (*p* > 0.05) in DMd were recorded between the substrate mixtures (0.518 and 0.466 for HC and HF, respectively; sem = 0.0231) or between the additives with either HC or HF, although the results recorded for this parameter supported numerically those of the gas production for both mixtures (Table 2). Similarly, no treatment differences were detected in microbial mass, either in the fraction associated with the liquid phase or in the solid phase (Table 3).

No treatment differences were recorded in the total VFA concentration, nor in the molar VFA proportions at 6 h incubation (Table 4), except for a higher (*p* = 0.022) proportion of BCFA with HF as substrate and a trend for the highest proportion of these when B was added to HF but not to HC (substrate–additive interaction, *p* = 0.076). At 12 h of incubation (Table 5), no treatment differences were detected in the total VFA concentration, but higher proportions of acetate (0.571 vs. 0.552; *p* = 0.032) and BCFA (0.0148 vs. 0.0128; *p* = 0.013) at the expense of butyrate (0.136 vs. 0.144; *p* = 0.028) were found in HF compared with HC. Between the substrates, the ammonia concentration was lower (*p* = 0.017) with HC at 6 h (5.19 vs. 5.93 mM; sem = 0.108), and differences tended towards significance (*p* = 0.050) at 12 h (7.25 vs. 7.93 mM; sem = 0.151). With HC as the substrate, the inclusion of B reduced the ammonia concentration compared to C at both 6 and 12 h incubation, but no differences between the additives were recorded with HF.

## 4. Discussion

Aiming to achieve highly productive results, intensive ruminant production systems are based on diets rich in ingredients with abundant nutrients that are fermented at a high rate. However, this strategy overloads the physiological buffering capacity of the rumen environment, promoting lower fibre degradation and increasing herd health problems related to acid–base disturbances. 

Several feed additives have been used to prevent these metabolic disorders [25,36], among which clays have been recommended to improve the physical qualities of feeds [9,12,13]. Moreover, researchers have indicated that these clays possess other benefits, such as antibacterial and detoxification properties, for livestock production [37,38] and can be used to promote digestion [21,37]. In terms of a comparison with results given in the literature, it is very important to take into account the variable nature of the different types of clay minerals available, which may influence their response to the rumen environment in different ways.

Among the tested clays, the addition of zeolite at 0.02 of the total substrate maintained a higher incubation pH than the unsupplemented substrates, to a greater extent with the substrate that included a higher concentrate proportion. This was mainly apparent with HC during the first 8 h of incubation (Figure 1), when the media were low-buffered and thus the potential of the additives was more clearly expressed. Thereafter, it was difficult to extrapolate firm conclusions, as the buffer concentration in the replaced medium from 8 h onwards may have masked the treatment behaviour. The results obtained in this work with natural zeolite agree with those observed in the in vivo study realized by Khachlouf et al. [9], who found that the inclusion of zeolite at a moderate level (less than 300 g/d) in the diets of dairy cows increased the ruminal pH, although this effect was reversed with a higher supplementation level (over 400 g/d). An increase in the ruminal pH with the addition of zeolites was also observed in other studies [5,6,7]. It has to be considered that most of these studies were conducted with high-concentrate diets. Yong et al. [39] indicated that zeolite has been shown to work as an alkalizer and have a great capacity for H^+^ exchange at different pH ranges. The moderate positive changes in the medium pH observed with zeolite when added to the HF substrate may be explained by the high proportion of alfalfa hay included in this mixture (0.65). Alfalfa hay, with a high intrinsic buffering capacity [40], provided a balanced fermentative environment [41] which potentially reduced the beneficial effect of zeolite observed in the high-concentrate diets. Similarly, no effect on ruminal pH was observed by Bosi et al. [11] when zeolite was added at 0.01 DM in dairy cow diets with a forage:concentrate ratio of 45:55, whereas Eng et al. [42] reported an increase in the rumen pH when adding 0.012 of zeolite to beef finishing diets. Bentonite affected the pH in the same sense, although to a lesser extent than zeolite, indicating that these minerals may maintain a balanced rumen pH, especially during starch fermentation, as was recorded by Fisher and Mackay [4] and Suzlburger et al. [6]. No relevant response in incubation pH was recorded by the inclusion of sepiolite, as was also observed by Ellitok and Guvlu [23] by adding 0.02 sepiolite in the concentrate of a diet offered ad libitum to lambs.

However, the more favorable environmental conditions promoted by zeolite and bentonite, especially with HC as the substrate, were not reflected in a clearly improved microbial fermentation process, as no major differences between Z and either C or F were observed in gas production (Table 2) or total VFA concentration (Table 4 and Table 5). Instead, in the case of HF, the volumes of gas produced with sepiolite or bentonite were lower than for the unsupplemented substrates in either the HC or HF incubations. In the case of sepiolite, this might be explained as a reduction in the activity of certain rumen bacterial species [23], thus causing a reduction in the fermentation rate. In contrast, Ivan et al. [24] reported an increase in the amount of VFA when sepiolite was included in a diet. In any case, the results obtained with HC and HF indicate that the diet composition affects the clay mineral effect, especially that of sepiolite and bentonite. These results are supported by those observed in the few previous studies that have been carried out with either sepiolite or bentonite [16,18,19]. 

In terms of the concentration and molar proportion of VFA, our findings indicate that, regardless of the type of diet, the inclusion of the three mineral clays in both diets had no effect on these parameters, partly because of a higher magnitude of the error term (coefficient of variation of 0.15 for VFA concentration vs. 0.04 for gas production at 12 h of incubation). These results are supported by those of Bosi et al. [11], Grabherr et al. [43], and Khachlouf et al. [9], especially when zeolite was used as an additive in lactating cow diets. However, Burçak and Yalçin [16] observed a linear increase in the rumen acetate proportion with up to 0.02 sepiolite, and McCollum and Galyean [44] reported an increase in the molar proportion of propionate when steers fed with high-concentrate diets were supplemented with 0.025 zeolite in their ration. It is worth considering that the clays in our experiment were included at 0.01 DM. The lack of differences in the molar proportions of VFA suggests that no major qualitative changes in microbial activity are promoted with these products according to their dosage amount, although no biodiversity studies were performed.

In agreement with our results, bentonite reduced the ammonia concentration in the unsupplemented substrate in HC, though not in HF. Previous studies [13,15,23,25] have indicated that the ammonia concentration in the rumen is reduced when bentonite or sepiolite are added as buffers in dairy cattle diets, and Khachlouf et al. [9] observed the same when zeolite was used as an additive. An increased ammonia concentration can be related to a high rumen protein degradation rate and, at a similar microbial concentration and activity level, it may suggest either that the addition of these mineral clays could reduce protein degradation or that clays have the ability to sequester released ammonia because of its high cation-exchange capacity. However, a lower ammonia concentration in the incubation media could also be partly justified by the sequestration of released amino acids from proteolysis; thus, Kihal et al. [45] reported the absorption indexes of lysine, methionine, and threonine from 0.31 to 0.62 with bentonite, sepiolite, or zeolite under in vitro conditions.

## 5. Conclusions

The present in vitro study confirms that supplementing zeolite as an additive in diets for ruminants has no major negative effects on ruminal fermentation; instead, it contributes to a more stable ruminal environment in terms of pH and ammonia concentration, especially in diets with high proportions of concentrate. However, this was not reflected in the extent of microbial fermentation. Regarding sepiolite and bentonite, both had a buffering response of a lower magnitude, but results for gas production suggest they reduce the extent of ruminal fermentation, depending on the type of diet. This indicates that the benefit of adding of these two clay minerals to the diets of ruminants is determined by the concentrate-to-forage proportion of the diet.

## Figures and Tables

**Figure 1 animals-12-00345-f001:**
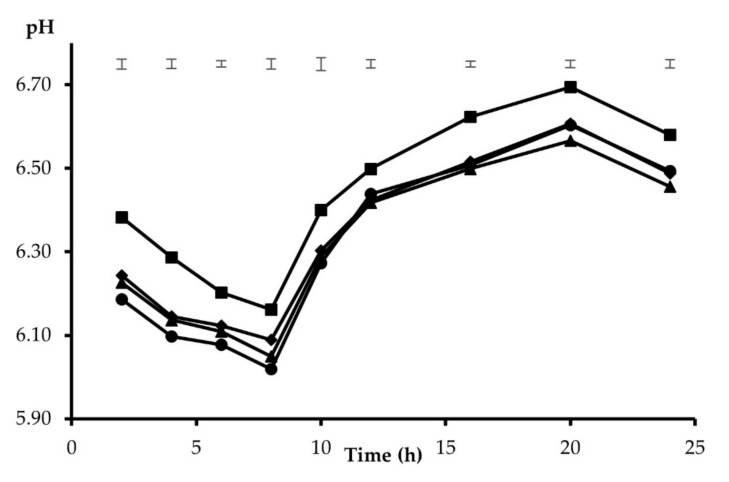
Pattern of pH changes of a 65:35 concentrate-to-forage (HC) substrate, incubated alone (C: ●) or supplemented with 10 mg/g of zeolite (Z: ■), bentonite (B: ◆), or sepiolite (S: ▲). Initial pH was 6.62 ± 0.16. Upper bars show standard error of means (*n* = 4).

**Figure 2 animals-12-00345-f002:**
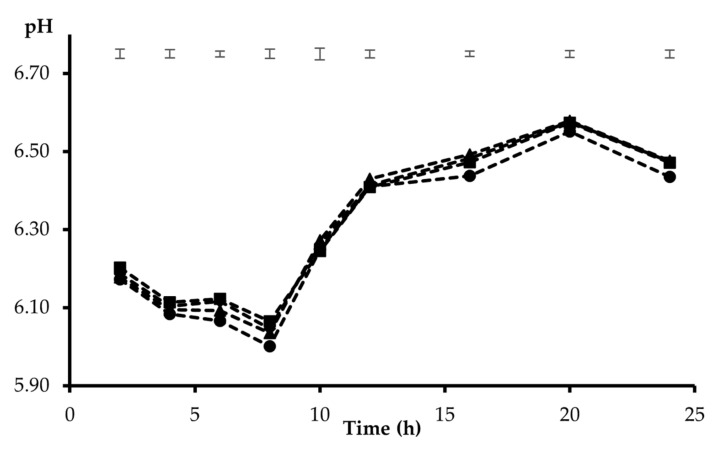
Pattern of pH change of a 35:65 concentrate-to-forage (HF) substrate, incubated alone (F: ●) or supplemented with 10 mg/g of zeolite (Z: ■), bentonite (B: ◆), or sepiolite (S: ▲). Initial pH was 6.62 ± 0.16. Upper bars show standard error of means (*n* = 4).

**Table 1 animals-12-00345-t001:** Chemical composition (g/kg dry matter) of ingredients and estimated composition of incubated substrates (high-concentrate, HC; high-forage, HF).

Ingredients	DM	OM	CP	NDF
Maize grain	859	987	85	103
Barley grain	899	977	89	178
Soybean meal	889	929	507	92
Wheat bran	861	942	195	484
Alfalfa hay	914	887	191	376
Estimated composition:				
HC	887	936	212	236
HF	899	913	202	301

DM, dry matter; OM, organic matter; CP, crude protein; NDF, neutral detergent fibre.

**Table 2 animals-12-00345-t002:** Gas production (mL/g organic matter, OM) from a 65:35 (HC) or a 35:65 (HF) concentrate-to-forage substrate, alone (C, F) or supplemented with 10 mg/g of clay minerals (zeolite, Z; bentonite, B; or sepiolite, S). sem: standard error of means.

	2 h	4 h	6 h	8 h	10 h	12 h	16 h	20 h	24 h
HC C	20.8 ^a^	37.6 ^a^	50.4 ^a^	61.4 ^a^	74.4 ^a^	86.9 ^a^	105.6 ^a^	124.4 ^a,b^	141.4 ^a^
Z	19.4 ^a^	33.2 ^a^	45.3 ^a^	56.0 ^a^	68.3 ^a^	81.7 ^a,b^	99.1 ^a^	116.9 ^b^	133.2 ^b^
B	19.8 ^a^	36.4 ^a^	49.3 ^a^	59.9 ^a^	73.2 ^a^	86.7 ^a^	105.5 ^a^	125.7 ^a^	141.6 ^a^
S	14.9 ^b,c^	28.3 ^b^	39.3 ^b^	47.9 ^b^	56.7 ^b^	65.6 ^c^	83.0 ^b^	100.7 ^c^	115.4 ^c^
HF F	17.6 ^a,b^	29.5 ^a,b^	39.1 ^b^	47.4 ^b^	58.0 ^b^	67.9 ^bc^	81.3 ^b^	95.6 ^c^	107.2 ^b,c^
Z	11.8 ^c^	23.1 ^b^	33.3 ^b,c^	41.8 ^b^	51.4 ^b^	60.8 ^c^	74.6 ^b^	88.4 ^c^	100.2 ^c^
B	10.9 ^c^	19.0 ^b^	27.7 ^c^	35.7 ^c^	44.4 ^c^	52.5 ^d^	64.7 ^c^	76.3 ^d^	86.2 ^d^
S	14.1 ^b,c^	26.2 ^b^	37.0 ^b^	45.7 ^b^	54.7 ^b^	64.1 ^c^	76.3 ^b^	89.5 ^c^	102.5 ^c^
sem	0.81	1.14	1.18	1.20	1.33	1.51	1.51	1.61	1.52
*p*-value									
Substrate	0.005	0.008	0.008	0.007	0.007	0.004	0.002	0.002	0.004
Additive	0.001	0.001	0.001	0.001	0.001	0.001	0.001	0.001	0.001
Subs.–Add.	0.001	0.001	0.001	0.001	0.001	0.001	0.001	0.001	0.001

^a–d^ Means in a column with different letters differ (*p* < 0.05). sem: standard error of means.

**Table 3 animals-12-00345-t003:** Dry matter disappearance (DMd) at 24 h incubation, liquid-associated microbial mass (LAM, µg/mL), and solid residue (SAM, µg/g) from a 65:35 (HC) or a 35:65 (HF) concentrate-to-forage substrate, alone (C, F) or supplemented with 10 mg/g of clay minerals (zeolite, Z; bentonite, B; or sepiolite, S). sem: standard error of means.

	DMd	LAM	SAM
HC C	0.506	330	290
Z	0.530	379	307
B	0.545	345	279
S	0.490	392	291
HF F	0.458	354	251
Z	0.471	365	272
B	0.460	374	264
S	0.474	421	300
sem	0.0175	35.68	26.60
*p*-value:			
Substrate	0.21	0.47	0.42
Additive	0.51	0.35	0.72
Subs.–Add.	0.30	0.93	0.79

**Table 4 animals-12-00345-t004:** Average total concentration (mM) and molar proportions (mmol/mmol) of volatile fatty acids (VFA) and ammonia concentration (mM) recorded at 6 h from a 65:35 (HC) or a 35:65 (HF) concentrate-to-forage substrate, alone (C, F) or supplemented with 10 mg/g of clay minerals (zeolite, Z; bentonite, B; or sepiolite, S).

	VFA	Acetate	Propionate	Butyrate	Valerate	BCFA	Ammonia
HC C	26.73	0.609	0.224	0.130	0.013	0.018	5.52 ^a^
Z	26.53	0.611	0.227	0.127	0.013	0.011	5.27 ^a,b^
B	25.14	0.614	0.225	0.128	0.017	0.011	4.90 ^b^
S	23.21	0.587	0.236	0.139	0.013	0.012	5.07 ^a,b^
HF F	25.03	0.615	0.219	0.128	0.013	0.013	5.88 ^a^
Z	26.93	0.618	0.219	0.124	0.013	0.013	5.82 ^a^
B	24.55	0.598	0.229	0.133	0.013	0.013	5.84 ^a^
S	27.13	0.618	0.219	0.125	0.013	0.013	6.16 ^a^
sem	1.200	0.0116	0.0062	0.0044	0.0005	0.0004	0.098
*p*-value							
Substrate	0.48	0.44	0.31	0.43	0.11	0.022	0.017
Additive	0.43	0.72	0.74	0.54	0.70	0.67	0.023
Subs.–Add.	0.14	0.29	0.42	0.22	0.20	0.076	0.005

^a,b^ Means in a column with different letters differ (*p* < 0.05). sem: standard error of means.

**Table 5 animals-12-00345-t005:** Average total concentration (mM) and molar proportions (mmol/mmol) of volatile fatty acids (VFA) and ammonia concentration (mM) recorded at 12 h from a 65:35 (HC) or a 35:65 (HF) concentrate-to-forage substrate, alone (C, F) or supplemented with 10 mg/g of clay minerals (zeolite, Z; bentonite, B; or sepiolite, S).

	VFA	Acetate	Propionate	Butyrate	Valerate	BCFA	Ammonia
HC C	25.37	0.562	0.245	0.142	0.024	0.013	8.03 ^a^
Z	22.26	0.546	0.260	0.145	0.024	0.012	6.86 ^b^
B	23.72	0.555	0.252	0.144	0.024	0.013	6.78 ^b^
S	23.10	0.545	0.259	0.147	0.024	0.013	7.34 ^a,b^
HF F	25.65	0.579	0.238	0.132	0.021	0.015	7.94 ^a,b^
Z	21.34	0.564	0.243	0.139	0.023	0.016	8.02 ^a,b^
B	21.98	0.562	0.247	0.139	0.023	0.015	8.00 ^a,b^
S	25.06	0.580	0.238	0.132	0.022	0.014	7.76 ^a,b^
sem	1.780	0.0136	0.0067	0.0052	0.0005	0.0007	0.217
*p*-value							
Substrate	0.55	0.032	0.13	0.028	0.16	0.013	0.050
Additive	0.24	0.68	0.54	0.76	0.79	0.62	0.054
Subs.–Add.	0.87	0.76	0.59	0.79	0.80	0.40	0.021

^a,b^ Means in a column with different letters differ (*p* < 0.05). sem: standard error of means.

## Data Availability

Data sharing is not applicable.

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
