# Peer review of "Rumen Fermentation of Feed Mixtures Supplemented with Clay Minerals in a Semicontinuous In Vitro System"

_animals, 2022, doi:10.3390/ani12030345_

Round 1

Reviewer 1 Report

The manuscript contained no novel information about clays, zeolites, bentonites, or sepiolite. To complete this work, the authors had to demonstrate an important point.

Please rewrite the title because the diets used were most likely not just for milking cows, but were also used in other ruminants.

Instead of stabilizing rumen fermentation, clays are used in ruminant diets to detoxify or bind mycotoxin. The levels you used in this work were commonly used for detoxification or differentiation. Please provide an estimate of the amount of clay used to inhibit mycotoxin in dairy diets.

Why was the pH in HC higher than in HF? Did you address this issue in the text??

You did present the gas productions as a graph, but it was difficult to see any statistical differences between treatments. Could you please enter those values into the table in the same manner as DMd in Table 3?

Because of interactions between substrate and additive, all mean values of ammonia concentrations should be superscripted (P <0.05).

Author Response

REVIEWER 1:

The manuscript contained no novel information about clays, zeolites, bentonites, or sepiolite. To complete this work, the authors had to demonstrate an important point.

In fact, our aim was to study the potential effect of clays on rumen fermentation, since most studies have been focused to their role as technical additives or mycotoxin binders. In contrast, the stabilization of microbial fermentation, that may depend on the rumen environmental conditions promoted by the type of diet (mainly the forage to concentrate ratio) has been seldom covered. This (the comparison of effects of different clays on rumen fermentation under either high concentrate or high forage feed ratios) is what we consider the novel approach of the study, and in fact our results support the adequacy of this hypothesis. A comment about this has been included in the Introduction section.

Please rewrite the title because the diets used were most likely not just for milking cows, but were also used in other ruminants.

The combination of feeds used to test the effect of clays were chosen as representative of high- and moderate- concentrate mixtures among the most commonly used diets for milking cows. Although these diets are not commonly used for intensive beef production or small ruminants feeding in our region of influence (Spain), we understand they can also be used for other ruminants under different productive situations. Consequently, the suggestion is accepted and the title has been corrected.

Instead of stabilizing rumen fermentation, clays are used in ruminant diets to detoxify or bind mycotoxin. The levels you used in this work were commonly used for detoxification or differentiation. Please provide an estimate of the amount of clay used to inhibit mycotoxin in dairy diets.

The levels used depend on the legal limitations for their inclusion in ruminant diets. In general, the level of clays used as mycotoxins binder ranges from 1 to 5 kg/t, although EFSA accepts additions up to 20 kg/t for bentonite, which is the unique clay registered as substance for reduction of feed contamination by mycotoxins (aflatoxin B1). A comment about this has now been included in the Introduction.

Why was the pH in HC higher than in HF? Did you address this issue in the text?

As this was an in vitro approach, the incubation pH was adjusted to a fixed level, the same for both mixtures, considering the usual pH range commonly found in vivo, but also allowing for a pH that would ensure the functioning of the in vitro methodology. Although incubation pH for the main effect HC resulted slightly higher (P<0.05) than in HF, differences were of minor magnitude (lower than 0.1 pH unit), as it is mentioned in the first paragraph of the Results section. Besides, the observed pH ranges can be considered as expectable in this type of diets and, as it is a buffered media, comparison between types of diets was not an objective of the paper, and are considered as representative of differences in responses among additives within concentrate to forage ratios.

You did present the gas productions as a graph, but it was difficult to see any statistical differences between treatments. Could you please enter those values into the table in the same manner as DMd in Table 3?

Figures 3 and 4 have been transformed into Table 2, and the subsequent tables have been renamed accordingly.

Because of interactions between substrate and additive, all mean values of ammonia concentrations should be superscripted (P <0.05).

This has been amended

Reviewer 2 Report

The research uses an in vitro methodology to evaluate the effects of some clays on rumen fermentations. Although not particularly innovative, the issue still has controversial aspects and for this reason the research is interesting from a scientific point of view.

The manuscript is well organized and described. The methodologies used are correct and well detailed. The experimental design is rigorous. Only some clarifications are necessary for some aspects of the incubation procedure and more caution is suggested in the conclusions.

Table 1. The protein content of the two diets is very high and fibre quite low. why this choice?

Figure 1. I suggest to indicate in the figure the pH at the beginning of the incubation.

M&M

L107 were used ankom bag for in vitro incubation? they don’t have a porosity of 45 mm; how the residue of fermentation was quantitatively transferred in Ankom bag for NDF analysis?

L109 Which medium was used? buffered with carbonate or phosphate?

Discussion: The effect of clay on pH is one of the few noted by the authors in the present work. The rapid reduction of pH at the beginning of incubation does not seem to be linked to the fermentation processes but rather to the solubilization of the diet and additives (Authors should comment this rapid drop of pH at the beginning of the incubation). The effect of the zeolite on the pH occurs at this time (from 0 to 2 h of incubation) and remains constant throughout the entire incubation. On the other hand, for the reader seems difficult to hypothesize that, compared to the control, 10 mg of clay can modify the pH of 80 mL of a strongly buffered culture medium and keep unchanged this effect for 24 h despite a flow of medium of 6.4 ml / h. Please comment this consideration and in case, I suggest caution in interpreting this result also because it’s not supported by other variations in the fermentation profile.

Line L251 “Addition of zeolite at 0,02”? or 0,01?

Conclusions: Similarly, the effects of clays on gas production kinetics are not supported by variations in the quantity and profile of VFAs. For this reason, the results of the present research do not support the hypothesis that clays “may reduce the extent of rumen fermentation” (see L315).

Simple summary and abstract: Please, revise the text in accordance with the previous considerations

Author Response

REVIEWER 2:

The research uses an in vitro methodology to evaluate the effects of some clays on rumen fermentations. Although not particularly innovative, the issue still has controversial aspects and for this reason the research is interesting from a scientific point of view.

Thanks for your comment. As we have answered to Reviewer 1, our aim was to study the potential effect of clays on rumen fermentation since most studies have been focused to other roles of clays, like as technical additives or mycotoxin binders. The stabilization of microbial fermentation conditions has not been commonly covered, and this is what we consider the novel approach of the study. A comment about this has been included in the Introduction

The manuscript is well organized and described. The methodologies used are correct and well detailed. The experimental design is rigorous. Only some clarifications are necessary for some aspects of the incubation procedure and more caution is suggested in the conclusions.

Table 1. The protein content of the two diets is very high and fibre quite low. why this choice?

The ingredient mixtures were planned according to those commonly used in practice, without having into account specific chemical composition. The reason why protein is quite high is the inclusion of alfalfa hay as forage source; at the moment of starting the incubation we do not have samples of other commonly used forages such as grass silage or corn silage, and we preferred using alfalfa hay instead of cereal straw as more representative of diets for milking cows. In any case, protein level does not have a major impact in microbial fermentation if N requirements are met, so rumen results can be extrapolated to practical conditions.

Figure 1. I suggest to indicate in the figure the pH at the beginning of the incubation.

Average initial pH (mentioned in text at the start of Results section) has been included in title of Figures 1 and 2.

L107 were used ankom bag for in vitro incubation? they don’t have a porosity of 45 mm; how the residue of fermentation was quantitatively transferred in Ankom bag for NDF analysis?

We did not used Ankom bags, but nylon bags with a checked pore size of 45 µm (the commercial source has now been mentioned to avoid confusions)

L109 Which medium was used? buffered with carbonate or phosphate?

As indicated, buffer used was that of Van Soest (consisting of 85% bicarbonate buffer and 15% phosphate buffer), as proposed by Theodorou et al. (1994) for their in vitro incubation system. what we did was to modify bicarbonate ion concentration along the incubation to allow for pH fluctuations, as it is explained and suggested by Amanzougarene and Fondevila (2018).

Discussion: The effect of clay on pH is one of the few noted by the authors in the present work. The rapid reduction of pH at the beginning of incubation does not seem to be linked to the fermentation processes but rather to the solubilization of the diet and additives (Authors should comment this rapid drop of pH at the beginning of the incubation). The effect of the zeolite on the pH occurs at this time (from 0 to 2 h of incubation) and remains constant throughout the entire incubation. On the other hand, for the reader seems difficult to hypothesize that, compared to the control, 10 mg of clay can modify the pH of 80 mL of a strongly buffered culture medium and keep unchanged this effect for 24 h despite a flow of medium of 6.4 ml / h. Please comment this consideration and in case, I suggest caution in interpreting this result also because it’s not supported by other variations in the fermentation profile

Our main objective was not to extrapolate results on pH pattern to what may occur in vivo, and this is why we centred in comparing treatments when medium is poorly buffered (first 8 h); instead, what we aimed is to create fermentation conditions similar to those occurring in vivo, in order to study the effect of clays on the rate and extent of microbial fermentation. Initial buffering level of the media in this incubation system was low, aiming to simulate what occurs in vivo (this was already discussed in a previous paper also published in this journal (Amanzougarene et al., 2020; Animals 10, 261). In such paper, it is concluded that this system allows for the study of acidification potential of dietary treatments at the time their fermentation is compared under more realistic conditions than closed batch systems. We decided not to go deeper in discussion about incubation conditions with this system as it has been previously covered in former papers (Fondevila and Pérez-Espés, 2008; Prates et al., 2010; Bertipaglia et al., 2010, Anim Feed Sci Technol 159, 88-95; Amanzougrene et al., 2020). Besides, it has to be considered that medium dilution occurs in all continuous and semicontinuous in vitro systems.

We consider that the initial drop of pH with zeolites shows its higher buffering capacity compared to the other treatments, that is appreciable when buffer concentration in the medium was low. Thereafter, the system allows for pH to recover by increasing buffer concentration in the replacement solution, simulating what occurs in vivo when the animal ruminates; however, along this part of the incubation period the high buffer concentration in the replacing solution may mask the effect of treatments, and this is why this aspect is not discussed. According to the suggestion from the reviewer, a comment about initial drop of pH has now been included (third paragraph of Discussion).

I disagree with the last comment from the reviewer: a 0.01 proportion of a buffering substance may be enough to promote a response in pH when the buffering capacity of the incubation solution is low (first 8 h). Such buffering effect has been also reported in literature. The flow of liquid phase does not noticeably affect presence and activity of clays, which are in the solid fraction. In fact, significant differences among treatments were detected in gas production and ammonia concentration, and these varied according to the concentrate to forage ratio. These aspects have been revised and the Discussion section has been modified accordingly for clarification.      

Line L251 “Addition of zeolite at 0,02”? or 0,01?

Sorry, but in my version the line mentioned (L251) does not match with the comment from the Reviewer about the level of zeolite addition. Anyway, all clays were included at 10 mg/g, that is, 0.01.

Conclusions: Similarly, the effects of clays on gas production kinetics are not supported by variations in the quantity and profile of VFAs. For this reason, the results of the present research do not support the hypothesis that clays “may reduce the extent of rumen fermentation” (see L315).

Many times, the detected differences in in vitro fermentation (gas production) are not manifested in differences in VFA concentration or in molar VFA proportions at fixed incubation times, probably because of a higher variability of these parameters. As an example, coefficients of variation of 0.15 for VFA concentration vs. 0.04 for gas production at 12 h incubation. In fact, though not significantly, VFA concentration at 12 h incubation was numerically lower for S in HC diet and for B in HF diet, in agreement with what was observed in terms of gas production. Therefore, it may be suggested that sepiolite (not all the clays tested) can reduce fermentation. This argument has now been included in Discussion.

Simple summary and abstract: Please, revise the text in accordance with the previous considerations

Simple summary and abstract have been revised according to these considerations.

Round 2

Reviewer 1 Report

The authors had responded to all comments.

Reviewer 2 Report

the authors answered and clarified most of the issues. for this reason I believe that the manuscript can be published in the present form